# TOKENIZED TRANSFORMER WORLD MODELS FOR CONTINUAL REINFORCEMENT LEARNING

## ABSTRACT

Catastrophic forgetting is a significant obstacle in continual reinforcement learning, as newly acquired skills overwrite earlier ones, resulting in sharp performance drops and hindering the transfer of knowledge to future tasks. Replay buffers and regularization can mitigate this drift, but often at the expense of brittle transfer, excessive computation, or policy instability. We address these limits with a tokenized, world-model–centric agent. A compact vector-quantized autoencoder discretizes frames into short token sequences; a Transformer world model predicts next-step tokens, rewards, and terminations. A task module fuses explicit task identifiers with trajectory-inferred context via feature-wise modulation, yielding task-aware yet shareable representations. Adaptation is localized by inserting low-rank adapters only into the world model (not the policy), thereby concentrating plasticity in dynamics while maintaining stable control. A heteroscedastic critic supplies uncertainty that gates an adaptive entropy bonus and prioritizes imagined rollouts; per-game moving-average reward normalization and absorbing-state rollouts further stabilize learning. On the six-game Atari CORA benchmark (Isolated Forgetting, Zero-shot Forward Transfer), the agent consistently exhibits lower forgetting with positive forward transfer in task-aware settings and reduced forgetting under equal interaction budgets in task-agnostic settings.

## 1 INTRODUCTION

Humans possess the remarkable ability to acquire new skills throughout life while retaining old ones and leveraging prior knowledge to accelerate future learning. By contrast, deep reinforcement learning (RL) agents typically excel in single-task training but often fail in *continual reinforcement learning (CRL)* settings, where tasks arrive sequentially and the agent must balance *plasticity* (learning new tasks quickly) with *stability* (retaining past knowledge) (Kirkpatrick et al., 2017; Rolnick et al., 2019; Powers et al., 2021). The dominant failure mode in CRL is *catastrophic forgetting*, where newly acquired skills overwrite previously learned ones, resulting in a sharp decline in performance. Overcoming this limitation is a prerequisite for deploying RL agents in real-world, non-stationary environments.

World models (Ha & Schmidhuber, 2018; Hafner et al., 2024; Agarwal et al., 2024) have emerged as a powerful paradigm to address sample inefficiency and support planning by learning a compact generative model of environment dynamics. Recent advances show that discrete, tokenized representations from vector-quantized autoencoders can improve compositionality and transfer across tasks (Yu et al., 2022). However, most world-model agents are still designed for fixed-task benchmarks, and their ability to scale to multi-task continual RL with forward transfer remains limited. Furthermore, while continual learning research has proposed strategies such as synaptic consolidation (Kirkpatrick et al., 2017), experience replay (Rolnick et al., 2019; Fedus et al., 2020), and environment design (Garcin et al., 2024), integrating these ideas with world models is underexplored.

In this work, we present a *tokenized transformer world model* architecture designed for continual reinforcement learning in multiple Atari games. Our system combines a tokenized visual representation, a causal World Model (WM) for sequence prediction, and an actor–critic policy trained on imagined rollouts. During online learning, we apply LoRA (Hu et al., 2021) to the WM (including attention and MLP blocks, as well as the observation-token head), while training the policy end-to-

end without adapters. This design preserves the pre-trained structure of the WM, localizes plasticity, and maintains policy optimization as simple and stable as possible.

We evaluate our method on the six-game Atari sequence proposed in CORA (Powers et al., 2021), measuring *Continual Evaluation*, *Isolated Forgetting*, and *Zero-Shot Forward Transfer*. Our results demonstrate that tokenized world models with adapter-based continual learning reduce forgetting and improve forward transfer compared to strong baselines, under fixed interaction budgets.

Our contributions can be summarised as follows:

- **Tokenized world-model agent.** We introduce a tokenized Transformer world model for continual RL that unifies VQ-VAE visual tokens, a causal world model for sequence prediction, and an actor–critic trained on imagined rollouts.

- **Localized plasticity via adapters.** We propose an adapter-based continual learning strategy that inserts LoRA only into the world model (not the policy), concentrating plasticity in dynamics while keeping control stable and simple.

- **One-sided shared task conditioning.** We design a FiLM-based (Perez et al., 2017) task module that blends explicit task identifiers with trajectory-inferred context and *updates it only through the world-model objective*. This preserves a single task interface for both stacks while preventing policy gradients from drifting the conditioner.

- **Two-stage pretrain → online alternation.** We pretrain the tokenizer and world model offline, then alternate real-only world-model updates (on adapters and heads) with actor–critic updates on imagined data online, improving data efficiency and stability under fixed interaction budgets.

## 2 RELATED WORK

**World models and model-based reinforcement learning.** World models (Ha & Schmidhuber, 2018; Hafner et al., 2024; Agarwal et al., 2024; Robine et al., 2023; Dedieu et al., 2025) learn a generative model of environment dynamics and have shown strong sample efficiency by enabling imagination-based training. Transformer-based world models, in particular, can capture long-term dependencies and outperform recurrent alternatives on Atari and open-ended benchmarks (Robine et al., 2023; Agarwal et al., 2024; Dedieu et al., 2025). Recent work further improves data efficiency through discrete tokenization (Yu et al., 2022; Agarwal et al., 2024), patch-based encoders (Dedieu et al., 2025), or active exploration (Kim et al., 2020). While these advances emphasize efficient single-task training, relatively few studies have explored their potential for continual multi-task reinforcement learning.

**Continual reinforcement learning.** Continual reinforcement learning (CRL) focuses on agents that learn a sequence of tasks without catastrophic forgetting (Kirkpatrick et al., 2017; Rolnick et al., 2019). Classical approaches include synaptic consolidation such as Elastic Weight Consolidation (EWC) (Kirkpatrick et al., 2017), parameter isolation (Rusu et al., 2022), and replay-based methods like CLEAR (Rolnick et al., 2019) and its extensions (Fedus et al., 2020). Benchmarking efforts such as CORA (Powers et al., 2021) formalize evaluation with metrics for continual evaluation, isolated forgetting, and zero-shot forward transfer, across task sequences in Atari, Procgen, MiniHack, and CHORES. Despite these advances, most continual RL methods remain model-free and have not leveraged the representational benefits of world models.

**Transfer and zero-shot generalization.** Transfer learning in RL aims to accelerate learning on new tasks using prior knowledge (Cheng et al., 2025; Garcin et al., 2024). Zero-shot transfer is particularly challenging, requiring agents to generalize without additional training. Data-regularized environment design (DRED) (Garcin et al., 2024) and large-scale pretraining of world-action models (Cheng et al., 2025) demonstrate that careful data design and joint optimization can improve forward transfer. Recent works also highlight the promise of leveraging pretrained large models for zero-shot or in-context RL in challenging games (Li et al., 2025). Our work differs in that it focuses on tokenized transformer world models with adapter-based continual learning, thereby bridging discrete representation learning with CRL benchmarks.

**Disentangled and tokenized representations.** Discrete and disentangled latent representations have been shown to facilitate modularity, compositionality, and interpretability in generative modeling (Yu et al., 2022). In the RL setting, discrete tokens serve as a compact and structured representation that can improve both policy learning and transfer (Agarwal et al., 2024). By integrating a VQ-VAE tokenizer with a transformer world model, we aim to exploit these benefits in the continual learning regime.

## 3 METHOD

We model control in a discrete visual space. At each time $t$, the environment yields an observation $o_t$. A tokenizer maps $o_t$ to tokens $z_t \in \{1, \ldots, V\}^K$. A task vector $c_t \in \mathbb{R}^d$ modulates embeddings via FiLM. A causal world model (WM) consumes the interleaved history $[\, z_{1:t}, a_{1:t} \,]$ and predicts $(\hat{z}_{t+1}, \hat{r}_t, \hat{d}_t)$ (Vaswani et al., 2023). Rollouts from the WM provide imagined trajectories that train an actor–critic (Hafner et al., 2024), which outputs a policy $\pi_\theta(a_t \mid z_{\leq t}, c_{\leq t})$ and a value $V_\psi(z_{\leq t}, c_{\leq t})$; the sampled $a_t$ closes the loop with the environment. Figure 1 illustrates the end-to-end data flow.

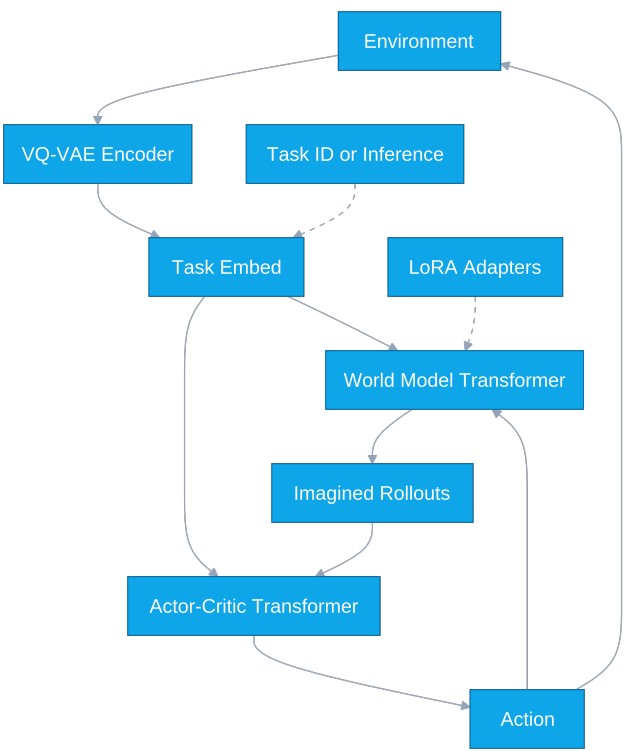

Figure 1: **System overview.** Environment $\rightarrow$ observation $o_t$. Tokenizer $\rightarrow$ tokens $z_t \in \{1, \ldots, V\}^K$. Task id or inference $\rightarrow$ task vector $c_t \in \mathbb{R}^d$. FiLM applies $c_t$ to observation/action embeddings (dashed arrows denote conditioning). The world model processes $[\, z_{1:t}, a_{1:t} \,]$ and outputs $(\hat{z}_{t+1}, \hat{r}_t, \hat{d}_t)$. During imagination, it unrolls $(\tilde{z}, \tilde{a}, \tilde{r}, \tilde{d})$ over a finite horizon conditioned on policy actions. The actor–critic consumes token features and $c_t$ to produce $\pi_\theta$ and $V_\psi$; the chosen action $a_t$ feeds back to the environment (solid arrows denote data flow).

### 3.1 IMPLEMENTATION DETAILS

**Visual tokenization with VQ-VAE.** Each $84\times84$ RGB frame is encoded by a VQ-VAE into $K$ discrete indices using a codebook of size $V$:

$$z_t = \text{VQ-VAE}(o_t) \in \{1, \ldots, V\}^K \tag{1}$$

Concretely, we obtain a $21{\times}21$ latent map, crop to $20{\times}20$, and partition it into a $4{\times}4$ grid of $5{\times}5$ non-overlapping patches to form $K{=}16$ tokens per frame. Commitment and codebook losses follow van den Oord et al. (2018); decoding to pixels is used only during VQ-VAE pretraining, never during imagined rollouts.

**Architectural causality and sequence layout.** The WM is causal end-to-end: given past [obs, act] tokens, it predicts the *next* frame's tokens (a $V$-way classification for each of the $K$ positions), the scalar reward, and a termination logit. The policy stack is two-stage: a *non-causal* per-frame encoder aggregates within-frame token sets, and a *causal* temporal Transformer operates across time.

**Task conditioning via FiLM.** Let $c_t \in \mathbb{R}^d$ be the task vector at time $t$; it blends a learned identifier $e_{\mathrm{id}}(g)$ (indexed by game id $g$ when provided) with an inferred descriptor from history $\tau_{1:t}$:

$$c_t \;=\; \alpha\, e_{\mathrm{id}}(g) \;+\; (1-\alpha)\, f_{\mathrm{inf}}(\tau_{1:t}), \quad \alpha \in [0,1] \tag{2}$$

Feature-wise linear modulation (FiLM) applies to both observation-token embeddings $x$ and (optionally) action embeddings $a$:

$$\mathrm{FiLM}(h; c_t) \;=\; s(c_t) \odot h + b(c_t), \quad h \in \{x, a\} \tag{3}$$

with $s(\cdot), b(\cdot)$ implemented by small MLPs (Perez et al., 2017).

**Action masking and actor temperature.** Atari exposes at most 18 discrete actions and each game uses a subset (the *minimal action set*). To keep a *single* policy head across all games—crucial in continual learning to avoid changing the output dimensionality and reinitializing weights—we fix the actor head to $|\mathcal{A}|{=}18$ logits. A per-game binary mask $M \in \{0,1\}^{|\mathcal{A}|}$ ($M_a{=}1$ iff action $a$ is legal in the current game) suppresses invalid actions by setting their logits to $-\infty$ *before* the softmax. We also use a learnable temperature $T \;=\; \exp(\mathrm{clip}(\log T, \log 0.5, \log 4.0))$ shared across games. Let $\ell \in \mathbb{R}^{|\mathcal{A}|}$ denote the raw actor logits; we form

$$\ell^{(T)} \;=\; \ell/T, \qquad \tilde{\ell}_a \;=\; \begin{cases} \ell_a^{(T)}, & M_a = 1, \\ -\infty, & M_a = 0, \end{cases} \qquad \pi(a \mid s) \;=\; \mathrm{softmax}(\tilde{\ell}). \tag{4}$$

This assigns exactly zero probability (and zero gradients) to illegal actions while retaining a single shared head across tasks; the temperature $T$ modulates exploration by uniformly scaling valid logits without affecting masked dimensions.

For entropy regularization and reporting, we normalize by the number of valid actions:

$$\bar{H} \;=\; \frac{\mathsf{H}(\pi)}{\log |\mathcal{A}_{\mathrm{valid}}|}, \quad |\mathcal{A}_{\mathrm{valid}}| \;=\; \sum_{i=1}^{|\mathcal{A}|} M_i \tag{5}$$

so that the target entropy is comparable across games with different action-set sizes.

**World model training.** Next-token prediction uses cross-entropy over $V$ classes for each of the $K$ positions. We use the Huber (smooth-$\ell_1$) loss on per-game normalized rewards:

$$\mathcal{L}_{\mathrm{rew}} = \frac{1}{|\mathcal{M}|} \sum_{t \in \mathcal{M}} \phi_\delta\big(r_t - \hat{r}_t\big) \tag{6}$$

with the piecewise penalty

$$\phi_\delta(u) = \begin{cases} \frac{1}{2} u^2 & |u| \leq \delta \\ \delta\big(|u| - \frac{1}{2}\delta\big) & |u| > \delta \end{cases} \tag{7}$$

where $r_t$ is the normalized target and $\hat{r}_t$ the prediction. Compared with MSE, Huber is *quadratic* for small residuals (preserving precision) and *linear* for large residuals (robust to outliers and heavy-tailed reward spikes), yielding more stable gradients under non-stationary reward scales. We always use a sigmoid focal loss (Lin et al., 2018) on the termination logit $\ell_t^{\mathrm{done}}$ with label $d_t \in \{0, 1\}$:

$$p_t \;=\; \sigma(\ell_t^{\mathrm{done}}), \mathcal{L}_{\mathrm{done}} \;=\; -\alpha\, d_t\, (1 - p_t)^\gamma \log p_t \;-\; (1 - \alpha)\,(1 - d_t)\, p_t^\gamma \log(1 - p_t) \tag{8}$$

Episodes contain many non-terminal steps ($d_t{=}0$), so plain BCE tends to be dominated by easy negatives. The focal modulator emphasizes *hard* (misclassified) steps and mitigates class imbalance, improving calibration around episode boundaries. Overall, the world-model loss is

$$\mathcal{L}_{\mathrm{WM}} = \mathcal{L}_{\mathrm{tok}} + \lambda_r\, \mathcal{L}_{\mathrm{rew}} + \lambda_d\, \mathcal{L}_{\mathrm{done}} \tag{9}$$

**Imagination with soft terminations.** Let $\hat{p}_{\mathrm{done},t}$ denote the WM's termination probability at step $t$. We define a *soft* per-step discount

$$\hat{\gamma}_t = \gamma_0\big(1 - \hat{p}_{\mathrm{done},t}\big) \quad \text{with } \gamma_0 \in (0,1) \tag{10}$$

which smoothly reduces credit assignment as the model anticipates termination (White, 2021). Imagined trajectories are *early-cut* when $\hat{p}_{\mathrm{done},t}$ exceeds a threshold $\tau_{\mathrm{stop}} \in (0,1)$ to avoid propagating learning signals far beyond predicted episode boundaries. Per-step losses on imagined data are weighted by

$$w_t = \prod_{k \le t} \hat{\gamma}_k \tag{11}$$

and both GAE (Schulman et al., 2018) and return targets are computed with the same $\hat{\gamma}_t$ for internal consistency. This scheme improves stability under uncertain terminations by interpolating between standard discounting and absorbing-state truncation.

**Policy learning on imagined rollouts.** The actor loss is a weighted advantage objective

$$\mathcal{L}_{\mathrm{actor}} = -\mathbb{E}\big[w_t \, \log \pi(a_t \,|\, s_t) \, \tilde{A}_t\big] \tag{12}$$

with normalized advantages

$$\tilde{A}_t = \mathrm{clip}\left(\frac{A_t - \mu}{\sigma}, -3, 3\right) \tag{13}$$

where $(\mu, \sigma)$ are batch statistics and $A_t$ is computed with $\hat{\gamma}_t$. The critic predicts $(\mu_V, \log \sigma_V^2)$ per state and is trained with a Gaussian NLL

$$\mathcal{L}_{\mathrm{critic}} = \tfrac{1}{2} \mathbb{E}\left[w_t \left(\frac{(R_t - \mu_V)^2}{\sigma_V^2} + \log \sigma_V^2 + \log(2\pi)\right)\right] \tag{14}$$

**Software and Libraries.** We interface environments through Gymnasium (Towers et al., 2024) with the Arcade Learning Environment backend (Machado et al., 2018). All Atari experiments follow the standard ALE protocol (sticky actions and the minimal action set). We accelerate Transformer attention with Flash-Attention kernels when available (Dao, 2024).

## 3.2 TRAINING PROCEDURE

**Pipeline overview.** Our pipeline has two stages. **Stage 1** performs *offline pretraining* in two sub-stages: (1a) a VQ-VAE tokenizer is trained to produce discrete visual tokens; (1b) a causal Transformer WM is pretrained on token/action sequences. **Stage 2** performs *continual online learning* with alternating policy updates and LoRA updates.

**Stage 1a: Tokenizer pretraining (offline).** We train a VQ-VAE on multi-game Atari replay to produce a discrete representation. Frames are resized to $84 \times 84$ RGB and tokenized with codebook size $V$ and $K$ tokens per frame. For each game in our benchmark set (the same six Atari tasks used in Stage 2), we collect a fixed budget of frames using a simple exploratory policy (random action with sticky probability and a minimal action set) and randomly sample windows for reconstruction.

Table 1: Trainable vs. frozen components during stage 1a (tokenizer pretraining)

| Component | Trainable? |
| --- | --- |
| VQ-VAE encoder / decoder / codebook | ✓ |

**Stage 1b: World model pretraining (offline).** Given the frozen tokenizer from Stage 1a, we pretrain a causal Transformer WM on short token/action trajectories of horizon $H_{\mathrm{pre}}$ sampled from the same multi-game replay. The WM jointly optimizes: (i) next-token prediction (cross-entropy over $V$ classes for each of the $K$ positions), (ii) reward regression (Huber on per-game normalized rewards), and (iii) termination prediction (sigmoid focal). We validate with token perplexity, reward RMSE, and precision/recall/F1 for the `done` head. This yields a task-conditioned dynamics model used to model real trajectories and generate imagined rollouts in Stage 2.

Table 2: Trainable vs. frozen components during stage 1b (WM pretraining)

| Component | Trainable? |
|---|---|
| VQ-VAE encoder / decoder / codebook | ✗ |
| WM Transformer (base weights) | ✓ |
| WM observation-token head | ✓ |
| WM reward head | ✓ |
| WM termination head | ✓ |
| Task embedding | ✓ |

**Stage 2: Online Continual Learning.** We alternate data collection, WM updates on real data, and policy updates on real/imagined data. We freeze all non-adapted WM parameters and train: (i) LoRA adapters attached to WM Transformer blocks, (ii) a LoRA-adapted observation-token head, and (iii) reward and termination heads directly (without LoRA). The policy backbone and heads are trained end-to-end without adapters. This localizes plasticity while preserving pretrained representations.

1. **Collect (real).** Roll out the current policy for horizon $H$ in the active game $g$ to gather real trajectories. Encode each frame with the tokenizer in Eq. (1) and store the resulting tokens (and cached encoder features, if used) in a *real-only* replay buffer with per-game bookkeeping. Append windows

$$(z_{t:t+H-1}, a_{t:t+H-1}, r_{t:t+H-1}, d_{t:t+H-1}, gid)$$

   to replay; these windows seed imagined rollouts in the AC-only phase and provide supervision for the WM-only updates

2. **WM-only (real).** Sample balanced real windows from replay and update the WM on *real* data only. Freeze the WM backbone and update LoRA adapters and prediction heads. Optimize the WM objective in Eq. (9), where the token term is next-token cross-entropy $\mathcal{L}_{\text{tok}}$, the reward term uses the Huber loss in Eq. (6) with penalty shape in Eq. (7), and the termination term uses the sigmoid focal loss in Eq. (8). This phase maintains modeling quality on the current task while constraining plasticity to a small parameter subspace.

3. **AC-only (imagined).** Sample starting tokens from replay (optionally restricted to the latest WM batch) and use the WM to generate imagined rollouts for $H$ steps. Compute returns and advantages with the soft discount defined in Eq. (10), and optimize the actor–critic with a normalized-entropy hinge toward target $H^\star$ using the normalization in Eq. (5). Weight per-step losses by the cumulative product $w_t$ from Eq. (11), which down-weights steps near predicted terminations. Terminate imagination early when the termination probability exceeds the threshold $\tau_{\text{stop}}$ and mask tokens/rewards beyond the cut.

Table 3: Trainable vs. frozen components during stage 2 (online)

| Component | Trainable? |
|---|---|
| Policy Transformer + actor/critic heads | ✓ |
| VQ-VAE encoder / decoder / codebook | ✗ |
| WM Transformer (base weights) | ✗ |
| WM Transformer (LoRA) | ✓ |
| WM observation-token head (LoRA) | ✓ |
| WM reward head | ✓ |
| WM termination head | ✓ |
| Task embedding | ✓ |

**Shared task conditioning with one-sided updates.** Both the WM and the policy consume the *same* task-conditioning network that produces the vector $c_t$ and modulates embeddings via FiLM in Eq. (3) (cf. the blend in Eq. (2)). During Stage 2 we update this network *only through the WM*

*objective* in Eq. (9); it is excluded from the policy optimizer. Thus, policy gradients do not modify the task-conditioning parameters, which localizes plasticity to the WM branch while preserving a single, consistent task interface across the two stacks.

# 4 EXPERIMENTS

## 4.1 RESULTS

Table 4 reports Atari CORA scores using the sign conventions: $F > 0$ quantifies performance *degradation* on past tasks (forgetting), $F < 0$ indicates *backward transfer*; $Z > 0$ indicates zero-shot improvement on future tasks, $Z < 0$ indicates harm. Under task-aware evaluation, our method attains $F = -0.08 \pm 0.01$, i.e., on average it exhibits backward transfer rather than degradation, reducing forgetting by an absolute 0.15 compared with the strongest baseline (CLEAR, $0.07 \pm 0.01$). Against regularization-based baselines (EWC $0.03 \pm 0.03$, Online EWC $0.16 \pm 0.01$, P&C $0.18 \pm 0.01$), the absolute reduction ranges from 0.11 to 0.26. For forward transfer, our method yields $Z = 0.02 \pm 0.01$, a small but consistent positive effect compared with baselines clustered around $0.00-0.01$.

In the task-agnostic setting, our method maintains $F = 0.00 \pm 0.01$, effectively preventing degradation on previously learned tasks and outperforming CLEAR by 0.07 and IMPALA by 0.23 in absolute terms. However, the zero-shot effect becomes negative without task information ($Z = -0.06 \pm 0.01$), in contrast to the slight positive transfer observed when task identity is available. Overall, the approach markedly suppresses cross-task interference and delivers consistent positive transfer when task cues are provided, while remaining competitive on the forgetting metric even in the agnostic regime.

Table 4: Atari CORA metrics. $F > 0$ denotes forgetting; $F < 0$ denotes backward transfer. $Z > 0$ denotes positive forward transfer; $Z < 0$ denotes negative forward transfer. Baseline results are reported by (Powers et al., 2021)

| Method | Avg $F$ | Avg $Z$ |
|---|---|---|
| IMPALA | $0.23 \pm 0.01$ | $0.01 \pm 0.00$ |
| EWC | $0.03 \pm 0.03$ | $0.01 \pm 0.02$ |
| Online EWC | $0.16 \pm 0.01$ | $0.00 \pm 0.00$ |
| P&C | $0.18 \pm 0.01$ | $0.00 \pm 0.01$ |
| CLEAR | $0.07 \pm 0.01$ | $0.00 \pm 0.00$ |
| **Ours (task-aware)** | **-0.08 $\pm$ 0.01** | **0.02 $\pm$ 0.01** |
| **Ours (task-agnostic)** | **0.00 $\pm$ 0.01** | **-0.06 $\pm$ 0.01** |

## 4.2 ANALYSIS

Following Powers et al. (2021), let $r_{i,j,\text{end}}$ be the expected return on task $T_i$ evaluated at the end of training task $T_j$, and let $r_{i,\text{all,max}} = \max_t |r_{i,t}|$ denote the maximum absolute return observed for $T_i$ over the run.

*Isolated Forgetting* quantifies how learning a later task $T_j$ affects a previous task $T_i$,

$$F_{i,j} = \frac{r_{i,j-1,\text{end}} - r_{i,j,\text{end}}}{|r_{i,\text{all,max}}|} \quad (i < j),$$

so $F_{i,j} > 0$ denotes forgetting and $F_{i,j} < 0$ denotes backward transfer.

*Zero-shot Forward Transfer* measures how learning an earlier task $T_j$ changes performance on a future task $T_i$ before training on it,

$$Z_{i,j} = \frac{r_{i,j,\text{end}} - r_{i,j-1,\text{end}}}{|r_{i,\text{all,max}}|} \quad (i > j),$$

so $Z_{i,j} > 0$ indicates positive forward transfer and $Z_{i,j} < 0$ negative transfer.

Because both $r_{i,j-1,\text{end}}$ and $r_{i,j,\text{end}}$ lie in $[-|r_{i,\text{all,max}}|,\ |r_{i,\text{all,max}}|]$, each metric is bounded by $F_{i,j}, Z_{i,j} \in [-2, 2]$, with 0 meaning no net change. For diagnosis, we also report row and column means computed only over valid pairs (upper triangle for $F$, lower triangle for $Z$), as well as the overall averages

$$\bar{F} = \frac{1}{|\{i < j\}|} \sum_{i<j} F_{i,j}, \qquad \bar{Z} = \frac{1}{|\{i > j\}|} \sum_{i>j} Z_{i,j}.$$

On Atari CORA, the headline averages (Table 4) are $\bar{F} = -0.08 \pm 0.01$ and $\bar{Z} = +0.02 \pm 0.01$ for the task-aware variant, indicating bounded interference with weakly positive transfer on average. The structure of $F$ by *next-trained* task shows that training *MsPacman* and *Krull* produces the strongest negative impact on earlier tasks (column means around $-0.28$ and $-0.17$), whereas training *Hero* is mildly beneficial on average (column mean about $+0.17$); *BeamRider* and *Star-Gunner* are close to neutral (approximately $-0.06$ and $-0.02$). The structure of $Z$ by *earlier* task highlights *SpaceInvaders* and *Hero* as strong positive sources (row means near $+0.23$ and $+0.15$), while *Krull* and *BeamRider* are weakly negative overall (about $-0.13$ and $-0.19$) and *StarGunner* is near neutral (about $-0.07$).

At the pair level, positive $F$-cells are sparse and small, whereas localized constructive effects (e.g., training *Hero* slightly improves prior *SpaceInvaders* with $+0.21$) and interference hot-spots (notably when the next task is *MsPacman*) delineate where coupling is most pronounced. Overall, these patterns suggest a few "hub" tasks (notably *SpaceInvaders* and *Hero*), whose representations transfer broadly, in contrast to tasks (*MsPacman* and *Krull*) that are more likely to induce interference when learned later.

### 4.3    ABLATION STUDY: EFFECT OF REMOVING LoRA FROM THE WORLD MODEL

**Setup.**    We compare the default system (world model + LoRA) against an ablated variant that *keeps the world model but removes LoRA during online learning*. All other components and evaluation protocols remain unchanged. We report task-aware isolated forgetting $F$ and zero-shot forward transfer $Z$ (Tables 5 and 7 for the main model; Tables 6 and 8 for the ablation), and also include the task-agnostic metrics (Tables 10 and 12). In each table, the *Avg* entry (last column) averages the corresponding off-diagonal entries.

**Forgetting.**    Under task-aware evaluation, the run with LoRA yields an average $\bar{F} = -0.08$ (Table 5), whereas removing LoRA yields $\bar{F} = -0.01$ (Table 6). Thus, removing LoRA *reduces the magnitude of backward transfer* by 0.07 (absolute), and forgetting remains non-positive in both settings. Column averages summarize how training each destination task $T_j$ affects previously learned tasks: more negative values indicate stronger backward transfer, while more positive values indicate greater forgetting. Without LoRA, training *BeamRider* produces the strongest backward transfer to prior tasks (column Avg $-0.17$ in Table 6); *MsPacman* shows a mild backward transfer (column Avg $-0.06$), and the remaining columns are near zero. The task-agnostic summary displays the same trend with an overall $\bar{F} = -0.04$ (Table 10), indicating no net forgetting on average.

**Forward transfer.**    With LoRA the task-aware average is $\bar{Z} = +0.02$ (Table 7); removing LoRA yields $\bar{Z} = -0.08$ (Table 8), a 0.10 absolute decrease with a sign flip to negative transfer. Row averages (last row) quantify the effect of each *source* task on future tasks prior to training them. Removing LoRA flips *SpaceInvaders* from positive to negative transfer ($+0.23 \rightarrow -0.26$) and amplifies the negative effect of *StarGunner* ($-0.07 \rightarrow -0.50$), while it turns *Krull* from negative to positive ($-0.13 \rightarrow +0.19$) and slightly strengthens *Hero*'s positive transfer ($+0.15 \rightarrow +0.20$); see Table 8. The task-agnostic aggregate is modestly negative, $\bar{Z} = -0.03$ (Table 12).

**Interpretation.**    LoRA introduces small, targeted adaptation subspaces inside the world model. This increases beneficial sharing across tasks, as reflected by higher $Z$ in the full model (Tables 7 vs. 8), at the cost of slightly greater interference (more negative $F$ by about 0.07; Tables 5 vs. 6). Removing LoRA weakens cross-task coupling: it reduces forgetting but also suppresses forward transfer. In practice, LoRA is advantageous when positive transfer/reuse is desired; disabling it is preferable when strict isolation between tasks is more important.

## 5 CONCLUSION

We introduced a tokenized, world–model–centric agent for continual reinforcement learning. The agent unifies discrete visual tokenization, a causal Transformer dynamics model, and an actor–critic trained on imagined rollouts. We localize plasticity by inserting LoRA adapters into the world model and conditioning behavior with FiLM, which injects task information while preserving policy stability. A heteroscedastic critic, per–task reward normalization, and termination–aware imagination further stabilize optimization.

Experiments on sequential tasks demonstrate reduced forgetting and consistent forward transfer compared to strong replay and regularization baselines, all under equal interaction budgets. Task cues amplify forward transfer; in their absence, interference remains controlled, but gains diminish. These results support the claim that concentrating adaptation in the dynamics model enables reuse without catastrophic interference.

Ablations corroborate the design. Removing adapters weakens cross–task reuse and lowers forward transfer, confirming that targeted adaptation in the world model is beneficial. Conversely, stricter isolation may be preferable when transfer risk outweighs reuse benefits.

Limitations include reliance on the quality of visual tokenization, modest sensitivity to task–cue availability, and computational overhead associated with imagination. Future work should scale tokenizers and horizons, strengthen task inference to reduce cue dependence, and explore adaptive or mixture–of–adapters to balance reuse and isolation per task pair. Evaluations beyond Atari–style domains will test robustness under longer sequences and distribution shifts.

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

# A CORA SCORE MATRICES

Table 5: Isolated forgetting $F$ (task-aware; rows: affected task $i$, columns: next-trained task $j$). **Setting:** world model *with* LoRA.

| Task | SpaceInvaders | Krull | BeamRider | Hero | StarGunner | MsPacman | Avg |
|------|---------------|-------|-----------|------|------------|----------|-----|
| SpaceInvaders | – | -0.17 | -0.05 | 0.21 | -0.13 | -0.42 | -0.11 |
| Krull | | – | -0.06 | 0.10 | 0.51 | -0.66 | -0.03 |
| BeamRider | | | – | 0.19 | -0.37 | -0.13 | -0.10 |
| Hero | | | | – | -0.08 | 0.08 | 0.00 |
| StarGunner | | | | | – | -0.25 | -0.25 |
| MsPacman | | | | | | – | – |
| **Avg** | – | -0.17 | -0.06 | 0.17 | -0.02 | -0.28 | **-0.08** |

Table 6: Isolated forgetting $F$ (task-aware; rows: affected task $i$, columns: next-trained task $j$). **Setting:** world model *without* LoRA.

| Task | SpaceInvaders | Krull | BeamRider | Hero | StarGunner | MsPacman | Avg |
|------|---------------|-------|-----------|------|------------|----------|-----|
| SpaceInvaders | – | 0.03 | -0.12 | 0.29 | 0.16 | -0.29 | 0.01 |
| Krull | | – | -0.21 | -0.09 | -0.02 | -0.13 | -0.11 |
| BeamRider | | | – | 0.07 | -0.07 | 0.05 | 0.02 |
| Hero | | | | – | 0.04 | -0.01 | 0.02 |
| StarGunner | | | | | – | 0.09 | 0.09 |
| MsPacman | | | | | | – | – |
| **Avg** | – | 0.03 | -0.17 | 0.09 | 0.03 | -0.06 | **-0.01** |

Table 7: Zero-shot forward transfer $Z$ (task-aware; rows: later task $i$, columns: earlier task $j$). **Setting:** world model *with* LoRA.

| Task | SpaceInvaders | Krull | BeamRider | Hero | StarGunner | MsPacman | Avg |
|------|---------------|-------|-----------|------|------------|----------|-----|
| SpaceInvaders | – | | | | | | – |
| Krull | 0.33 | – | | | | | 0.33 |
| BeamRider | -0.02 | 0.12 | – | | | | 0.05 |
| Hero | 0.96 | -1.00 | 0.00 | – | | | -0.01 |
| StarGunner | -0.18 | 0.33 | -0.56 | 0.25 | – | | -0.04 |
| MsPacman | 0.04 | 0.04 | 0.00 | 0.06 | -0.07 | – | 0.01 |
| **Avg** | 0.23 | -0.13 | -0.19 | 0.15 | -0.07 | – | **0.02** |

Table 8: Zero-shot forward transfer $Z$ (task-aware; rows: later task $i$, columns: earlier task $j$). **Setting:** world model *without* LoRA.

| Task | SpaceInvaders | Krull | BeamRider | Hero | StarGunner | MsPacman | Avg |
|------|---------------|-------|-----------|------|------------|----------|-----|
| SpaceInvaders | – | | | | | | – |
| Krull | 0.01 | – | | | | | 0.01 |
| BeamRider | -0.36 | -0.08 | – | | | | -0.22 |
| Hero | -0.26 | 0.55 | -0.77 | – | | | -0.16 |
| StarGunner | -0.35 | 0.26 | 0.18 | -0.06 | – | | 0.01 |
| MsPacman | -0.35 | 0.02 | 0.02 | 0.46 | -0.50 | – | -0.07 |
| **Avg** | -0.26 | 0.19 | -0.19 | 0.20 | -0.50 | – | **-0.08** |

Table 9: Isolated forgetting $F$ (task-agnostic; rows: affected task $i$, columns: next-trained task $j$). **Setting:** world model *with* LoRA.

| Task | SpaceInvaders | Krull | BeamRider | Hero | StarGunner | MsPacman | Avg |
|---|---|---|---|---|---|---|---|
| SpaceInvaders | – | -0.14 | 0.34 | 0.06 | -0.41 | 0.07 | -0.02 |
| Krull | | – | -0.23 | 0.40 | 0.49 | -0.54 | 0.03 |
| BeamRider | | | – | 0.32 | -0.16 | -0.39 | -0.08 |
| Hero | | | | – | -0.04 | 0.04 | 0.00 |
| StarGunner | | | | | – | 0.15 | 0.15 |
| MsPacman | | | | | | – | – |
| **Avg** | – | -0.14 | 0.06 | 0.26 | -0.03 | -0.13 | **0.00** |

Table 10: Isolated forgetting $F$ (task-agnostic; rows: affected task $i$, columns: next-trained task $j$). **Setting:** world model *without* LoRA.

| Task | SpaceInvaders | Krull | BeamRider | Hero | StarGunner | MsPacman | Avg |
|---|---|---|---|---|---|---|---|
| SpaceInvaders | – | -0.03 | 0.36 | -0.21 | 0.12 | -0.36 | -0.02 |
| Krull | | – | -0.32 | -0.01 | -0.07 | -0.15 | -0.14 |
| BeamRider | | | – | 0.01 | -0.21 | 0.15 | -0.02 |
| Hero | | | | – | 0.02 | 0.00 | 0.01 |
| StarGunner | | | | | – | 0.05 | 0.05 |
| MsPacman | | | | | | – | – |
| **Avg** | – | -0.03 | 0.02 | -0.07 | -0.04 | -0.06 | **-0.04** |

Table 11: Zero-shot forward transfer $Z$ (task-agnostic; rows: later task $i$, columns: earlier task $j$). **Setting:** world model *with* LoRA.

| Task | SpaceInvaders | Krull | BeamRider | Hero | StarGunner | MsPacman | Avg |
|---|---|---|---|---|---|---|---|
| SpaceInvaders | – | | | | | | – |
| Krull | 0.12 | – | | | | | 0.12 |
| BeamRider | -0.05 | -0.16 | – | | | | -0.11 |
| Hero | 0.90 | -1.00 | 0.00 | – | | | -0.03 |
| StarGunner | -0.59 | 0.40 | -0.38 | 0.06 | – | | -0.13 |
| MsPacman | -0.33 | 0.09 | 0.03 | 0.13 | -0.18 | – | -0.05 |
| **Avg** | 0.01 | -0.17 | -0.12 | 0.10 | -0.18 | – | **-0.06** |

Table 12: Zero-shot forward transfer $Z$ (task-agnostic; rows: later task $i$, columns: earlier task $j$). **Setting:** world model *without* LoRA.

| Task | SpaceInvaders | Krull | BeamRider | Hero | StarGunner | MsPacman | Avg |
|---|---|---|---|---|---|---|---|
| SpaceInvaders | – | | | | | | – |
| Krull | 0.12 | – | | | | | 0.12 |
| BeamRider | -0.14 | -0.25 | – | | | | -0.20 |
| Hero | 0.55 | -0.51 | -0.22 | – | | | -0.06 |
| StarGunner | -0.38 | 0.35 | -0.08 | -0.25 | – | | -0.09 |
| MsPacman | 0.09 | 0.29 | -0.32 | 0.13 | 0.13 | – | 0.06 |
| **Avg** | 0.05 | -0.03 | -0.21 | -0.06 | 0.13 | – | **-0.03** |

# B HYPERPARAMETERS

Table 13: Model hyperparameters.

| Hyperparameter | Symbol | Value |
|---|---|---|
| **Tokenizer** | | |
| Input resolution | — | $84 \times 84$ RGB |
| Tokens per frame | $K$ | 16 ($4{\times}4$ grid of $5 \times 5$ patches) |
| Codebook size | $V$ | 512 |
| Token / feature dimension | $d$ | 512 |
| EMA decay / commitment | — | 0.8 / 0.1 |
| **Task conditioning** | | |
| Known tasks | $G$ | 6 |
| Action vocabulary | $|\mathcal{A}|$ | 18 |
| Task / token feature dim | $d$ | 512 / 512 |
| **World Model** | | |
| Model width | $d$ | 512 |
| Attention heads | $h$ | 8 |
| Feed-forward dim | — | 2048 |
| Layers | $L$ | 6 |
| Activation / norm | — | GeLU / Pre-LN |
| Dropout | $p$ | 0.2 |
| **Policy** | | |
| Per-frame encoder (non-causal) layers | $L_f$ | 2 (Transformer; $d{=}512$, $h{=}8$, FF $= 2048$) |
| Temporal stack (causal) layers | $L_t$ | 6 (Transformer; $d{=}512$, $h{=}8$, FF $= 2048$) |
| Pooling | — | Learnable query attention pooling |
| Actor head | — | Linear $\rightarrow |\mathcal{A}|$ |
| Critic head | — | Linear $\rightarrow 2$ (value mean $\mu_V$, log-variance $\log \sigma_V^2$) |

Table 14: Training hyperparameters.

| Hyperparameter | Symbol | Value |
|---|---|---|
| **Pretraining: Tokenizer** | | |
| Optimizer | — | AdamW (Loshchilov & Hutter, 2019) |
| LR / weight decay (enc+dec) | — | $3\times10^{-4}$ / $1\times10^{-4}$ |
| LR / weight decay (codebook) | — | $1\times10^{-4}$ / 0 |
| Batch size / workers / epochs | $B$ | 256 / 16 / 100 |
| Loss | — | MSE (pixel reconstruction) |
| **Pretraining: World Model** | | |
| Trajectory horizon | $H_{\mathrm{pre}}$ | 8 |
| Optimizer (betas) | — | AdamW (0.9, 0.95) |
| LR by groups | — | Trunk+TaskCond: $1\times10^{-4}$; Obs-head: $1\times10^{-4}$; Reward/Done heads: $1\times10^{-5}$ |
| Weight decay | — | Trunk+Obs-head: $10^{-2}$; others: 0 |
| Batch size / workers | $B$ | 512 / 8 |
| AMP / grad clip | — | bfloat16 / 5.0 |
| **Online: Policy & WM adapters** | | |
| Environment | — | Frameskip 4; sticky 0.25; minimal action set |
| Imagination horizon | $H$ | 15; early stop if $p_{\mathrm{done}} > 0.9$ |
| Discount / GAE | $\gamma_0$, $\lambda$ | $\hat{\gamma}_t = 0.99(1 - p_{\mathrm{done},t})$; $\lambda = 0.95$ |
| Loss weights | $\lambda_r$, $\lambda_d$ | 10, 50 |
| Batch sizes (WM / policy) | $B_{\mathrm{wm}}$, $B_\pi$ | 32 / 128 |
| Optimizer (policy) | — | AdamW (0.9, 0.95) |
| LR / weight decay (policy) | — | Trunk: $1\times10^{-4}$ / $3\times10^{-4}$; Heads&Temp: $1\times10^{-4}$ / 0 |
| Scheduler (policy) | — | Cosine, $T_{\mathrm{max}}$=200 |
| Optimizer (WM online) | — | AdamW (0.9, 0.95) |
| LR / weight decay (WM online) | — | Trunk: $1\times10^{-4}$ / $10^{-2}$; Obs-head: $1\times10^{-4}$ / $10^{-2}$; Reward/Done: $1\times10^{-5}$ / 0; TaskCond: $1\times10^{-4}$ / 0 |
| AMP / grad clip | — | bfloat16 / 1.0 |
| Entropy regularization | $\eta$, $H^\star$ | $\eta$=0.02 (hinge) to $H^\star$=0.30 on $\mathsf{H}(\pi)/\log|\mathcal{A}_{\mathrm{valid}}|$ |
| Critic loss | — | Gaussian NLL with predicted $\log \sigma_V^2$ (clamped) |

## C  PSEUDO-CODE FOR PRETRAINING AND ONLINE TRAINING

---

**Algorithm 1** Pretraining

---

**Require:** Replay $\mathcal{D}$ of $(x_t, a_t, r_t, d_t)$; tokenizer parameters $\theta_{\text{tok}}$; world model parameters $\theta_{\text{wm}}$; window length $H_{\text{wm}}$

1: **function** PRETRAINTOKENIZER($\mathcal{D}, \theta_{\text{tok}}$)
2:     **while** not converged **do**
3:         Sample batch of frames $x$ from $\mathcal{D}$
4:         Encode $x \to$ token features; vector-quantize to codes; decode to $\hat{x}$
5:         Update $\theta_{\text{tok}}$ to minimize reconstruction loss
6:     **end while**
7: **end function**

8: **function** PRETRAINWORLDMODEL($\mathcal{D}, \theta_{\text{wm}}, \theta_{\text{tok}}$)
9:     **while** not converged **do**
10:         Sample windows of length $H_{\text{pre}}+1$ from $\mathcal{D}$
11:         Tokenize frames with tokenizer; keep $(a_t, r_t, d_t)$
12:         Predict next tokens, reward, and termination for each step
13:         Compute $\mathcal{L}_{\text{obs}} + \mathcal{L}_{\text{rew}} + \mathcal{L}_{\text{done}}$
14:         Update $\theta_{\text{wm}}$ by minimizing total loss
15:     **end while**
16: **end function**

---

**Algorithm 2** Online Training (Policy + World Model Adapters)

---

**Require:** Vectorized environments; pretrained tokenizer and world model; imagination horizon $H_{\text{imag}}$
1: Initialize policy parameters; attach low-rank adapters to the world model
2: **while** training **do**
3:     **Collect real data:** run policy with action masks; store (tokens, $a, r, d$) in replay
4:     **Update world model on real:** sample windows from replay; update only adapters and heads
5:     **Imagine rollouts:**
        Start from replay states; for $t=0{:}H-1$:
         sample $a_t$ from policy; predict next tokens, reward, done; stop a branch if $p_{\text{done}}$ is high
6:     **Compute returns/advantages:** use discounted weights with termination-aware discount
7:     **Update policy:** policy-gradient loss + value loss + entropy target penalty
8:     Periodically evaluate and save the best checkpoint
9: **end while**

---