# OpenReview forum: "Tokenized Transformer World Models for Continual Reinforcement Learning"
_ICLR.cc/2026/Conference — ICLR 2026 Conference Withdrawn Submission_

### Official Review · Reviewer_DaZn · 2025-10-27

**Soundness:** 1
**Presentation:** 1
**Contribution:** 2
**Rating:** 2
**Confidence:** 4

**Summary:**

The paper targets limitations of replay- and regularization-based continual RL methods—namely brittle transfer, high computational overhead, and policy instability. It proposes a tokenized, world-model–centric agent: an autoencoder discretizes image frames into token sequences, a Transformer predicts next-step tokens, rewards, and terminations, and a task module infers task identity from trajectory features. The world model is adapted via LoRA, while the critic employs adaptive entropy gating and prioritizes imagined rollouts. Evaluated on the Atari CORA benchmark, the method reports improved performance over baselines.

**Strengths:**

1.	The integration of discrete tokenization, a predictive world model, task identification, and LoRA-based adaptation is a novel combination that is well motivated for continual RL.

**Weaknesses:**

1.	The approach presumes access to task identity (either known a priori or learned). Given this, why not assign a task-specific output head per task rather than fix the policy head to |A|=18? A fixed shared head becomes increasingly inflexible as the number of tasks (and effective action outputs) grows.
2.	The choice of temperature clipping bounds $log T \in [\log 0.5, \log 4.0]$ lacks justification. Moreover, the objective term $L_{\text{tok}}$ is referenced but not clearly defined; its role and weighting within the overall loss should be made explicit.
3.	Although the problem is framed as online continual learning, the method relies on offline pretraining on Atari datasets. This raises concerns about information leakage and comparability to methods trained strictly online. The paper should clarify data usage, freeze/update protocols during pretraining vs. online phases, and ensure apples-to-apples comparisons.
4.	Several compared baselines lack proper citation and may be outdated for an ICLR 2026 submission. Parameter fairness is also unclear: Do baselines receive analogous pretraining or comparable model capacity? Report parameter counts, pretraining status, and training budgets to ensure fair comparison.
5.	The method comprises multiple interacting components (discrete autoencoder, world model, task identifier, LoRA adaptation, entropy gating, rollout prioritization). The ablation study is superficial and does not quantify each component’s contribution. A thorough study should include component-wise removal and hyperparameters.
6.	The overall pipeline is difficult to follow, and the current experimental comparisons appear imbalanced, which undermines confidence in robustness. A clearer methodological description (with pseudocode) and stronger fairness controls would improve credibility.
7.	And how the proposed methods address the limitations described in the Abstract (brittle transfer, excessive computation, policy instability)?

**Questions:**

See the weakness.

---

### Official Review · Reviewer_uUo8 · 2025-10-29

**Soundness:** 2
**Presentation:** 1
**Contribution:** 2
**Rating:** 4
**Confidence:** 4

**Summary:**

The paper proposes a continual reinforcement learning method which integrates VQ-VAE, Transformer world model, and imagination-based actor–critic.

**Strengths:**

Discrete representation with VQ-VAE and two-stage training enhance compositionality and sample efficiency.

Action masking makes this method suitable for various tasks.

**Weaknesses:**

This paper has a limited benchmark scope, more tasks, such as Procgen, MiniHack, CHORES, D4RL, and Continual World, will help show the effectiveness of this method.

More baselines will benefit to illustrate the superiority of this method. Please refer to the questions for more detailed comments.

A richer presentation of experiments is important. Currently, this paper only presents experiments in tabular form.

This paper needs an elegant paper structure diagram.

**Questions:**

1. At line 174, the authors did not introduce what $f_{inf}(\tau_{1:t})$ is.
2. In line 185, how is T selected?
3. According to the current calculation method, the probabilities of legal actions in the policy in Equation 4 are all the same, and the calculation of the current action's probability distribution seems to have no relation to the policy network.
4. Equation 5 does not introduce $H(\pi)$.
5. How is $\delta$ selected in Equation 7?
6. In line 213, I suggest the authors add references related to BCE.
7. Equation 8 does not introduce the meaning of gamma, and Equation 9 does not introduce the coefficients used to balance the loss.
8. What is gid in Stage 2?
9. Does t represent the reinforcement learning time step or the number of model updates?
10. For CRL using world models, there are the following related methods, and I suggest the authors compare against the following related methods
[1] Continual Offline Reinforcement Learning via Diffusion-based Dual Generative Replay
[2] t-DGR: A Trajectory-Based Deep Generative Replay Method for Continual Learning in Decision Making
[3] Continual Reinforcement Learning via Diffusion-based Trajectory Replay
11. The environments selected in this paper use the same action space configuration, for example, all using Atari series environments, where the action spaces have predefined sets, with the maximum environment action set being 18 actions. However, in reality, the action spaces of different environments may vary greatly, such as in MuJoCo robot environments, where the action spaces differ significantly due to different robot types. Furthermore, how does this paper address the continual reinforcement learning problem with continuous action spaces? If the state space of the environment changes (which is quite common in practical scenarios), how do the authors handle this?
12. Do all tasks share the VQ-VAE codebook, and are the encoder and decoder task-specific for different tasks?
13. Why did the authors choose to use LoRA for training? Atari tasks are relatively simple—what I mean is, why not train a small network separately for each task? This would eliminate the need for action masks, VQ-VAE encoder and decoder, and world model parameters, and would also remove the requirement for the first-stage pretraining on offline datasets.

---

### Official Review · Reviewer_k2VZ · 2025-10-31

**Soundness:** 2
**Presentation:** 3
**Contribution:** 2
**Rating:** 2
**Confidence:** 4

**Summary:**

In this paper, the author proposes a tokenized, Transformer-based world-model agent for continual reinforcement learning (CRL) on Atari. This pipeline encodes frames with a VQ‑VAE into discrete tokens, trains a causal Transformer “world model” to predict next tokens, reward, and termination, and trains an actor–critic from imagined rollouts. For CRL, the paper adds LoRA adapters only in the world model (policy has no adapters), and uses a FiLM task-conditioning module combining explicit game IDs and trajectory-inferred context but updates it only through the world-model loss, and introduces several stabilization details including heteroscedastic critic, rollout priority, per‑game reward normalization and action masking. On CORA–Atari (six-game sequence), the method reports reduced forgetting and small positive zero-shot forward transfer in task-aware settings and minimal forgetting in task-agnostic settings.

**Strengths:**

* This paper is written clearly and present in an organized way.
* This paper targets at an important topic in reinforcement learning, continual learning world models.
* This paper presents a clear engineering of a World model-based continual reinforcement learning agent. The paper assembles components that are individually well-motivated: discrete tokenization (VQ‑VAE), a causal Transformer world model, imagination-based policy learning, and PEFT via LoRA.

**Weaknesses:**

**Limited methodological novelty w.r.t. existing tokenized Transformer WMs for Atari.**
VQ‑VAE tokenization + Transformer world model + imagined rollouts—has already been developed and evaluated in previous work (DART, “Learning to Play Atari in a World of Tokens”; IRIS, Transformers are Sample-Efficient World Models); world models like Dreamer series also underpin imagination-based policy learning across diverse domains. LoRA adapters only in the WM and the “one‑sided” FiLM conditioner is more like an incremental re-assembly of known ingredients rather than a new algorithmic idea or theory.


**Evaluation protocol mismatch with CORA baselines**
The paper says Atari experiments follow the standard ALE protocol with sticky actions and the minimal action set and compares directly to baseline numbers reported in CORA. However, CORA’s published Atari results use deterministic environments without sticky actions. Therefore, copying baseline numbers from CORA (CLEAR, EWC, Online‑EWC, P&C) and comparing them with new results obtained under different environment dynamics and action-space constraints is not an apples‑to‑apples comparison.

**Missing strong and natural baselines.**
The baseline set mirrors classic CRL (CLEAR, EWC/Online‑EWC, P&C), but omits modern world‑model baselines particularly relevant here:  DreamerV2/DreamerV3/DreamerV4 adapted to the same CRL schedule, DART/IRIS (Transformer WM), Continual‑Dreamer (a WM‑based CRL agent with experience selection), CSR (Causality-guided Self-adaptive Representation-based WM). Without these, it is difficult to attribute gains to the proposed design rather than to the general advantages of using a tokenized Transformer WM.

**Effect sizes are small and statistical evidence is unclear.**
The average zero-shot forward transfer reported in task-aware settings is 0.02 ± 0.01, which is marginal and may not be practically meaningful. As it stands, it is hard to judge whether the positive Z is robust rather than a noise-level fluctuation. Why authors do not consider report the return in the new environments at the same time?

**Questions:**

Please refer to the weakness section for my concerns and suggestions.

---

### Official Review · Reviewer_TNm3 · 2025-10-31

**Soundness:** 3
**Presentation:** 3
**Contribution:** 2
**Rating:** 4
**Confidence:** 2

**Summary:**

This paper proposes a **tokenized transformer world model** (TTWM) for continual reinforcement learning (CRL), combining VQ-VAE visual tokenization, LoRA-based localized adaptation, FiLM-based task conditioning, and imagination-driven policy learning. The results on the Atari CORA benchmark show reduced forgetting and modest forward transfer compared to baselines such as CLEAR and EWC.

**Strengths:**

* The combination of discrete tokenization, Transformer-based dynamics, and LoRA-based localized plasticity is an interesting and elegant approach to continual RL.
* The idea of constraining adaptation to the world model rather than the policy is well-motivated and may improve stability and interpretability.
* The technical description (especially of the loss functions, causal layout, and training procedure) is detailed and reproducible.

**Weaknesses:**

* All experiments are conducted on a six-game Atari CORA sequence. There is no evaluation on continuous-control or non-visual tasks, which weakens claims of generality.
* Forward transfer ($Z = 0.02 \pm 0.01$) and backward transfer ($F = -0.08 \pm 0.01$) improvements, while consistent, are marginal. It’s unclear if they are statistically or practically significant.
* The only ablation studies LoRA removal. Other components (FiLM conditioning, reward normalization, heteroscedastic critic, or absorbing rollouts) are not tested independently.
* The authors claim “equal interaction budgets” but do not report computational costs, number of parameters, or rollout ratios (real vs. imagined). These are critical for fair comparison with model-free baselines.
* The paper reports negative forward transfer without task cues, yet claims the system is “competitive.” It remains unclear how task inference operates in this case, and whether the model collapses to average behavior.
* No visualizations or qualitative examples are shown (e.g., token reconstructions, trajectory rollouts, or adapter activations), which could substantiate claims about interpretability or compositionality.

**Questions:**

Q: Why were adapters inserted only into the world model?

Q: Did you try symmetric LoRA in both the world model and policy?

Q: How do you ensure that freezing the policy backbone does not hinder adaptation to tasks with substantially different action dynamics?

Q: The Transformer has 6 layers with $d=512$, but each frame produces only 16 tokens. Could the bottleneck be too small to capture diverse Atari dynamics?

Q: Are the CORA baselines re-implemented or taken directly from Powers et al. (2021)?

Q: Why does the model produce positive transfer only when task ids are given? Is the inference module too weak?

Q: The paper introduces several heuristics (Huber reward loss, sigmoid focal done loss, soft terminations). Which of these contribute most to training stability?

Q: Would the proposed architecture scale to continuous control or non-visual CRL tasks (e.g., MuJoCo or DMControl)?

---

### Note · Authors · 2025-11-19

I have read and agree with the venue's withdrawal policy on behalf of myself and my co-authors.